# Microstructure and Mechanical Properties of Severely Deformed Polypropylene in ECAE (Equal Channel Angular Extrusion) via Routes A and C

**DOI:** 10.3390/polym14235287

**Published:** 2022-12-03

**Authors:** Qifeng Jiang, Ramdane Boulahia, Fahmi Zaïri, Iurii Vozniak, Zhengwei Qu, Jean-Michel Gloaguen, Xiaobing Liu

**Affiliations:** 1Key Laboratory of Fluid and Power Machinery, Xihua University, Chengdu, 610039, China; 2Laboratory of Advanced Mechanics, University of Sciences and Technology Houari Boumediene, Algiers 16111, Algeria; 3Laboratoire de Génie Civil et géo-Environnement, Université de Lille, IMT Nord Europe, JUNIA, Université d’Artois, ULR 4515-LGCgE, 59000 Lille, France; 4Centre of Molecular and Macromolecular Studies, Polish Academy of Sciences, 90363 Lodz, Poland; 5International School of Business Management and Technology, 59100 Roubaix, France; 6Unité Matériaux et Transformations, Université de Lille, CNRS, INRAE, Centrale Lille, UMR 8207-UMET, 59000 Lille, France

**Keywords:** severe plastic deformation, equal channel angular extrusion, polypropylene, microstructure, mechanical properties

## Abstract

Equal channel angular extrusion (ECAE) is a solid-state extrusion process for modifying microstructures via severe plastic deformation without modifying the specimen cross section. In this study, changes in the microstructure and mechanical properties of polypropylene resulting from extrusion orientation route A (no rotation between extrusions) and extrusion orientation route C (a rotation of 180° between extrusions) are investigated using a 90° die-angle tooling outfitted with back pressure. Important differences are reported for the ECAE-induced deformation behavior between the two processing routes. A focus is made on the occurrence of heterogeneous plastic deformations (periodic shear banding and warping) for both routes and the control and inhibition of the plastic instabilities via regulated back pressure and ram velocity. Wide-angle X-ray scattering is carried out to characterize the structural evolution as a function of the processing conditions including route, extrusion velocity and BP application. The mechanical properties of the specimens machined from the ECAE pieces are examined under different loading paths including uniaxial tension/compression and simple shear. Full-field displacements converted to volumetric strains revealed the profound impacts of the processing route on the deformation mechanisms during tensile deformation.

## 1. Introduction

Equal channel angular extrusion (ECAE) is an ingenious solid-state extrusion process that deforms materials to a large simple shear strain without modifying the specimen cross section. The general principle of the process is shown schematically in Figure 1 in the L-configuration. The tool is a block with two intersecting channels of equal cross section. The specimen is placed in the vertical channel and then forced to pass, in the solid state, into the horizontal channel under the action of a ram that moves vertically. The extruded specimen is thus sheared in the crossing plane of the two channels with high degrees of plastic deformations inducing substantial alterations in microstructure and mechanical properties. In contrast to other solid-state forming processes (e.g., forging, ram-extrusion, or die-drawing) inducing a continuous reduction of the specimen cross section, the ECAE process preserves the specimen cross section and can be then repeated to induce very large cumulative plastic deformations in the extruded material. Furthermore, by changing the shear plane orientation by re-orienting the specimen between successive extrusions, very different microstructures can be generated. As illustrated in Figure 1, two fundamental ECAE routes can be employed: extrusion orientation route A in which the specimen is re-extruded in the same orientation as the previous pass and extrusion orientation route C in which the specimen is rotated by 180° and then re-extruded.

The first works concerned only metals but were gradually extended to polymers due to the high potential of the ECAE method in controlling microstructure and properties. Numerical simulations have matured to a point where they are widely employed for recommendations of the optimum conditions ensuring the homogeneity of plastic strain distribution and then of microstructures in ECAE-processed polymers [1,2,3,4,5,6,7]. Nonetheless, a reliable prediction of the intrinsic material behavior in connection to the alteration in the yielding mechanisms is a prerequisite to provide efficient tool design guidelines. The development of constitutive models for solid polymers is a very active research area that is constantly enriched by accounting for the complex connection among time-dependent nonlinear large-strain deformation, external parameters (such as temperature, strain rate and pressure), structural factors (including crystallinity index, crystal size, molecular weight, entanglement, cross-linking, etc.) and deformation mechanisms (such as crystallographic texturing and strain-induced preferred orientations of the amorphous molecular network) (see references [8,9] and the state-of-the-art of the literature given in the introductions of these recent papers). In order to take advantage of the potential of the ECAE process, the numerical analyses must be complemented by experimental efforts in specific investigations since it is not possible to provide a general rule. Optimizing the processing parameters requires a specific study of each particular material to quantify the mechanics of the process as well as the changes in the microstructure and mechanical properties of the extruded material.

Experimentally, the ECAE effectiveness in altering the morphology and the mechanical properties was reported in the literature for a variety of semi-crystalline and glassy amorphous polymers, such as polycarbonate [1,6,10,11,12,13], polymethylmethacrylate [12,13,14], low-density polyethylene [15,16], high-density polyethylene [16,17,18,19,20,21], polyoxymethylene [18], polypropylene (PP) [17,22,23,24,25,26,27,28,29], polyamide-6 [16,19,30,31], polyethylene terephthalate [32,33,34,35], ultra-high molecular weight polyethylene [36], acrylonitrile butadiene styrene [35], polytetrafluoroethylene [16,18,19], polylactide [37,38], fiber-reinforced polyacetal composites [39], polyamide-6-clay nanocomposites [30,40,41] and graphite-carbon-reinforced ultra-high molecular weight polyethylene composites [42,43]. The reader may also refer to overviews of the state-of-the-art of the literature [44,45,46]. The ECAE-induced changes in the microstructure and mechanical properties are mainly related to three inter-dependent factors: (i) initial material structural factors and intrinsic material properties, (ii) tool geometry (e.g., corner angle) and (iii) processing variables (e.g., friction, extrusion velocity, extrusion temperature, number of extrusion sequences and processing route). All these factors may change the deformation of the extruded piece from the expected homogeneous simple shear to a more complex heterogeneous field.

In a previous work [29], we examined how the ECAE process of PP works for producing high plastic deformation when repeated eight times and its impact on tensile mechanical properties in relation to crystalline texturing. The present companion paper is particularly focused on the mechanism of the occurrence of heterogeneous plastic deformations (periodic shear banding and warping) for PP specimens extruded twice via the processing routes A and C, and is even more focused on the control and inhibition of the instabilities via regulated back pressure (BP) and ram velocity. After using angular extrusion tests to analyze macroscopic flow behavior, the changes in microstructure and mechanical properties are investigated under a variety of conventional test conditions using different geometries and loadings, namely uniaxial tension/compression and simple shear. The changes in deformation mechanisms (plastic dilatation vs. shear yielding) due to both routes are shown during tensile deformation thanks to a digital image correlation system for local strain measurements.

## 2. Material and Methods

### 2.1. Material and Specimens

A commercial PP homopolymer was provided by the Goodfellow Company (Huntingdon, UK) in the form of 10 mm thick compression-molded plates: density *ρ* ≈ 0.95 g/cm^3^, weight-average molar weight *M_w_* ≈ 180,000 g/mol, number-average molar weight *M_n_* ≈ 25,000 g/mol and crystal content *χ*_c_ ≈ 55%. The 10 mm thick plates were cut into 10 mm × 75 mm long pieces for ECAE processing. Although no preferential orientation is expected, prismatic pieces were machined along the same direction as the as-received PP plates. The PP pieces were surfaced on the cutting facets and were polished.

### 2.2. ECAE Experiments

A stainless steel ECAE die was designed to be easily set up on an electromechanical Instron-5800 testing machine (Instron Precision Ltd., Sandwich, Kent, UK) with a capacity of 30 kN and an appropriate load cell. The tool is characterized by a square channel cross section of 10 mm × 10 mm and an internal angle of 90° between the two channels. ECAE channels and PP pieces have the same cross-sectional dimensions but with a clearance fit. Curved outer and inner angles are employed in the intersection area of the two channels in order to improve the flow capability of the material. An outer corner angle of 10° and an inner radius of 2 mm are used. BP can be applied in opposition to the flow direction in order to enhance the filling status of the extruded piece upon ECAE, especially in the intersection area. Before each extrusion, the die was lubricated using silicone grease. PP pieces were extruded twice under two processing scenarios: route A (no rotation of the ECAE-processed piece at the second pass) and route C (a rotation of 180° at the second pass). Extrusions were conducted at room temperature under constant ram velocities (4.5 to 45 mm/min), either without BP or with BP kept constant at 900 N. The load and the ram displacement were recorded for each extrusion.

### 2.3. Mechanical Tests and Specimens

A variety of mechanical tests, including simple shear, uniaxial compression and uniaxial tension, were used to examine the mechanical properties. The mechanical tests were performed under an initial equivalent strain rate of 0.001/s (under constant cross-head velocity conditions) at room temperature.

#### 2.3.1. Specimens

For each kind of test, specimens were machined from the ECAE-processed PP pieces. The specimen shape and dimensions are given in Figure 2.

For the uniaxial tensile tests, specimens with a 12.5 mm gauge length were prepared (the tensile axis was in the ECAE flow direction). Parallelepiped specimens of 20 mm × 5 mm × 5 mm were used for the simple shear tests. The cross-sectional area value was assigned to prevent flexure effects. For uniaxial compression tests, cubic specimens of 6 mm side were machined. A side-to-side ratio of one was chosen in order to prevent buckling. Care was taken to ensure the specimen ends were smooth and parallel in order to minimize the incidence of specimen shearing during the uniaxial compression. In addition, the platens of the testing machine were lubricated using silicone grease to reduce friction with the specimens as much as possible.

#### 2.3.2. Mechanical Tests

The mechanical tests were carried out on an electro-pulse Instron-5500 testing machine equipped with suitable testing rigs. The simple shear tests were carried out using a U-bolt and a hook which symmetrically shear two sections of the specimen, as described elsewhere [47,48]. The shear stress is calculated from the measured force and the area of two sheared sections, and the shear strain corresponds to the shear angle. In order to obtain proper information regarding the deformation distribution in the PP specimens during the uniaxial loading paths, the tensile tests were performed with the help of an accurate optical strain measuring technique based on digital image correlation. It is based upon correlating the gray levels of each image of the specimen surface in the deformed state with their counterpart of the image of the specimen surface in the undeformed state. The measurements require an artificial random and non-overlapped speckle pattern generated by (very thin) dots sprayed with an airbrush filled with Indian ink in order to obtain randomized gray-level distributions. The speckle diameter was as uniform and small as possible to ensure a good spatial resolution. A CCD camera (Imager E-lite) (LaVision GmbH, Göttingen, Germany) was placed in front of the sample and captured during the tensile loading of the random speckle pattern illuminated by a strong white light beam. Images were registered at regular intervals and digitized in 1024 × 1024 pixels. The analysis was performed with DaVis software developed by LaVision (LaVision GmbH, Göttingen, Germany). The zone of interest of each image was divided into small square sub-images of 24 × 24 pixels. The displacement vector was determined using the corresponding sub-image pairs extracted from the undeformed and deformed states. By achieving the analysis on numerous sub-images, the full-field displacements were obtained. The precision of displacement measurement is about 0.05 pixels. Typical error on the deformation is about 0.0005.

### 2.4. WAXS Measurements

Wide-angle X-ray scattering (WAXS) patterns were recorded at room temperature in order to quantify the texture modifications after ECAE processing. Slices were cut from the ECAE-processed pieces by careful diamond sawing. With the X-ray beam along the flow direction of the ECAE-processed pieces, the experiments were carried out using the Ni-filtered Cu-Kα radiation from a 2 kW Panalytical sealed tube operated on an INEL 2000 generator (INEL, Barcelona, Spain) at 40 kV and 20 mA. The 2D-WAXS patterns were recorded on a Photonic Science VHR CCD camera (PHOTONIC SCIENCE, Saint Leonards-on-sea, UK). Corrections were applied for background scattering, geometry and intensity distortions of the detector.

## 3. Results and Discussion

### 3.1. Deformation Behavior

The experimental results of the angular extrusion tests in terms of pressing load and deformation behavior are first examined. Figure 3 presents the evolution of the pressing load as a function of the ram displacement upon the first pass extrusion for the different ECAE processing conditions. Photographs of the PP pieces at the end of the ECAE operation are also provided in the figure.

Different stages referring to this figure can be highlighted. The first linear stage corresponds to the beginning of the pressing until the peak where the piece starts to cross the intersection area and the local plastic strain begins to develop. The load then starts to decrease to a minimal value due to the plastic zone growth until it diffuses over the piece width. A hardening which causes the load to increase can be noticed beyond the minimum value with a slope strongly dependent on the processing parameters. The latter induces profound modifications to the way PP behaves during the ECAE process. Without BP, plastic instabilities along the piece length may be observed (as also reported by Boulahia et al. [25]) in ECAE-processed PP. They are characterized by periodic waves on the top surface of the piece accompanied by stress drop (observed in the pressing load evolution) along with an alternation of un-sheared bands (opaque) and sheared bands (translucent) tilted at roughly 45° from the flow direction. The occurrence of these periodic plastic instabilities may be associated with a stick-slip phenomenon due to an unstable balance between PP/die contact friction force and PP yield strength, as manifested by the periodic load oscillations. The ram velocity has a significant effect on the plastic deformation along the ECAE-processed piece that appears more homogeneous at the lowest ram velocity with the disappearance of the shear-banding behavior. The use of BP also promotes the homogeneity of the plastic flow along the longitudinal direction, except for the edges exhibiting a low deformation because only a portion is slightly sheared at the beginning and the end of the extrusion. Slight inhomogeneous deformations are still observed at the highest ram velocity.

For the two processing routes, respectively, route A and route C, the pressing load is plotted in Figure 4 and Figure 5.

The sudden load increase, corresponding to the onset of piece shearing, is shifted due to the piece length reduction subsequent to the previous extrusion. With the use of BP, the load magnitude is increased due to the pressure dependency of PP yield strength and the increase in PP/die contact friction force. It can be also observed that the pressing load decreases when no BP is applied. Figure 4 and Figure 5 present photographs of the PP pieces that were removed from the die in order to appreciate the piece warping. The material displays completely different plastic flows when comparing both routes. Whereas the alternating bands are suppressed in route A, it leads to the highest warping. A significant reduction of the warping is obtained by using route C, especially when BP is applied. The extent of the warping may be due to the existence of the residual stress and concurrent stress relaxation processes in the PP pieces. It is believed that it leads to a local plastic deformation gradient from the top surface to the bottom surface of the piece (a lesser-sheared zone at the bottom surface is obtained since the inner part of the piece flows faster than the outer part [2]). In other words, the deformation mechanism in the bottom part is bending rather than shearing. In contrast to the other processing conditions for which they did not appear, the PP piece processed via route C without BP still shows periodic plastic instabilities at the highest ram velocity.

### 3.2. Crystallographic Texturing

Figure 3, Figure 4 and Figure 5 present the 2D patterns recorded along the transverse direction (in the translucent band, if any) after the first pass extrusion and the two processing routes. The figures give a general overview of the plastic mechanisms of the process, especially regarding the destruction of the PP initial spherulitic structure provoked by severe deformation. The latter gives a reduction of the light scattering that gives the translucent appearance of the sheared bands. The initial light-diffusing superstructure was kept in the opaque bands since they were not sheared as confirmed by the recorded 2D patterns (not shown here) exhibiting perfect crystalline isotropy. The WAXS patterns with the X-ray beam along the flow direction (not shown here) did not reveal any crystalline orientation and were roughly isotropic (whatever the processing conditions). By contrast, the WAXS patterns with the X-ray beam along the transverse direction exhibit a strong diagonal texturing characterized by a strong diagonal reinforcement of the three first reflections. They correspond to a preferred orientation of the (110), (040) and (130) planes containing the chain axis parallel to the shear direction, which is consistent with the expected shearing mechanism [49]. When BP is applied at the highest ram velocity, scattering reinforcements on quadrant positions of the inner reflections are observed as the manifestation of a double crystalline texturing of the (hk0) planes with a tilted orientation of the chain axis with respect to the shear plane. Double crystalline texturing can arise from either the occurrence of a double population of oriented crystallites or from a twinning mechanism. A more uniform texturing along the extruded specimen is also obtained when the extrusion velocity is decreased.

Different plastic processes in the crystalline phase are obtained when comparing both routes due to the change in the shear plane orientation, a tilting of the active crystallographic shear plane in route A and a reverse in route C. Quadrant (hk0) reflections symmetric with regard to the shear plane are observed in route C. The strong pressure sensitivity of PP plastic yielding leads to a modification of the plastic processes when BP is applied. The ram velocity reduction results in uniform texturing, even without BP.

### 3.3. Mechanical Properties

The stress-strain response during simple shear and uniaxial compression loading paths is given in Figure 6. Compressive strain and stress are plotted as positive values. The as-received PP material behavior is also plotted for both loading paths for comparison purposes.

The initial linear elastic behavior is followed by a rollover to yield but with a completely different plastic flow for the two loading paths. For the as-received PP material, the initial flow stress is significantly lower in simple shear than in uniaxial compression. However, after extrusion, the compressive yield strength becomes comparable to the shear yield strength. Nonetheless, the strain-hardening region is different for these two loading paths; nearly zero strain hardening can be observed in simple shear and progressive strain hardening can be observed in uniaxial compression [49]. This difference is related to distinct microstructure changes with deformation between simple shear [50] and uniaxial compression, inducing a planar molecular orientation process. The as-received PP material and the extruded specimens show the same plastic flow in simple shear. In compression, the initial elastic stiffness is observed to exhibit a negligible dependence on processing parameters. By contrast, they affect both the initial flow stress and the plastic flow with the appearance of a more or less pronounced strain softening.

The inhomogeneity of mechanical properties in extruded pieces along the longitudinal direction was also characterized. Figure 6b,c show a significant difference in compressive strength between a specimen containing three successive translucent-opaque-translucent (TOT) bands and another containing three successive opaque-translucent-opaque (OTO) bands. To highlight the inhomogeneity of mechanical properties across the piece width, the tensile stress-strain responses of two specimens extracted from the middle and bottom parts of the extruded piece are presented in Figure 7. The top part was not evaluated due to the presence of periodic waves on the top surface of the piece. A difference between the middle and the bottom can be observed after ECAE processing. When the PP piece is processed via route A for the second extrusion, a strong heterogeneity of the tensile flow behavior is obtained. By contrast, route C leads to more homogeneous mechanical properties across the piece width.

Figure 8, Figure 9 and Figure 10 display the effect of processing parameters on the tensile stress-strain response. The highly nonlinear tensile response of the as-received PP material can be clearly observed. It is characterized by strain softening after yielding followed by very moderate strain hardening until the final break. After the first ECAE pass, the strain softening extent decreases significantly, whereas the strain hardening ability increases, and the processing parameters have considerable effects on the hardening rate and the stretchability. The strong effects of the processing parameters observed after one extrusion diminish considerably when PP is processed via route C for the second pass, including the strain at break. Due to the dependency of the extrusion velocity on the specific crystallographic configuration and the maximum value of the resolved shear stress, the yield strength decreases with the velocity increase. The velocity at which the second extrusion is performed in route A alters the BP effect on the stress level. The latter is strongly increased for the highest ram velocity, and it is considerably attenuated for the lowest ram velocity.

### 3.4. Tensile Deformation Mechanisms

In order to give more information on the deformation pattern of the tensile specimen, a detailed analysis of the local deformation was also carried out. Examples of the deformed specimens and the strain field distribution are provided in Figure 8, Figure 9 and Figure 10 for a given cross-head displacement. The (true axial) strain field is shown in the color level scale. It can be observed that the local strain is not uniform across the specimen length with a significant effect of the processing route on the strain localization. For more homogeneous ECAE-processed specimens, a very diffuse strain instability is obtained. When the ECAE-processed specimen displayed the alternating bands, multiple strain localizations were developed. The sheared bands exhibit specific crystallographic configuration (texturing with their (hk0) planes tilted at roughly 45° from the flow direction), and the plastic shear instability is more likely to occur in these zones when they are stretched along the flow direction. After strain instabilities occur in the sheared bands, the un-sheared bands plastically deform gradually with continued stretching. It can be noted that multiple strain localizations are also observed, although the specimen appeared roughly homogeneous. More interestingly, it can be observed that, although the strain distribution depends on the processing factors when the specimen is extruded in route C, the overall stretchability is roughly invariable.

The full-field displacements obtained during tensile deformation were converted to volumetric strains. Figure 11 presents the plastic dilatation response (calculated by eliminating the elastic Poisson’s ratio effect) of the as-received PP material after ECAE processing via route A and route C. The tensile flow behavior is also provided in the figure for each condition. The plastic volumetric strain, for which the onset coincides with the nonlinearity of the stress-strain curve, is an indicator of the damage accumulation during tensile deformation manifested by the development of voids nucleated within the rubbery amorphous phase. As a visual confirmation of the presence of micro-voids (with a higher size than the light wavelength, typically 0.6 μm), whitening of the as-received PP specimen was observed. When PP is processed via route A, the plastic volumetric strain also constitutes an important part of the apparent deformation (the voiding process occurring between the crystalline blocks formed after the two successive extrusions). When compared with the as-received specimen, an equivalent voiding damage magnitude is obtained. Nonetheless, no whitening was observed in the extruded specimen, indicating the presence of smaller void sizes. It can be observed that the processing route plays a significant role in the deformation mechanisms of PP. Indeed, when PP is processed via route C, the plastic volumetric strain becomes negligible with regard to the applied strain level. The double shear orientation imposed on the crystalline blocks in route C seems to prohibit the voiding process during subsequent tensile deformation, the shear yielding becoming thus the dominant mechanism of plastic deformation.

## 4. Concluding Remarks

ECAE deformation of PP was examined using an L-configuration facility after one pass and two passes via route A and route C under different extrusion velocities with and without the application of BP. Particular attention was given to the global and local evaluation of the PP response. Several conclusions may be drawn:The homogeneous simple shear expected undergone by the extruded piece was found to be a heterogeneous deformation manifested by periodic shear banding and warping. Both pressing force and deformation heterogeneity were observed to be strongly dependent on the processing route in terms of BP and ram velocity. The results obtained by route A showed significant warping. The flow localization persisted in route C but significantly lower warping was observed; the results via this route become more homogeneous with the application of BP.Microstructural observations using WAXS experiments gave useful information about the evolution of the crystalline microstructure showing how different crystalline textures developed according to the processing parameters. Texturing changed from single-shear to twin-like shear orientation in the shear direction.The flow behavior was examined with the help of shear tests, compression tests and tension tests of specimens machined from the ECAE pieces. Important changes in tensile properties are observed between the first pass and the second pass with a significant effect of the processing route on the yield strength and the post-yield response.The tensile deformation mechanisms were altered by the processing route passing from a dominant cavitation mechanism for route A to a dominant shear yielding mechanism for route C.

The next step will be to combine the experimental investigations of PP extrusion with numerical simulations.

## Figures and Tables

**Figure 1 polymers-14-05287-f001:**
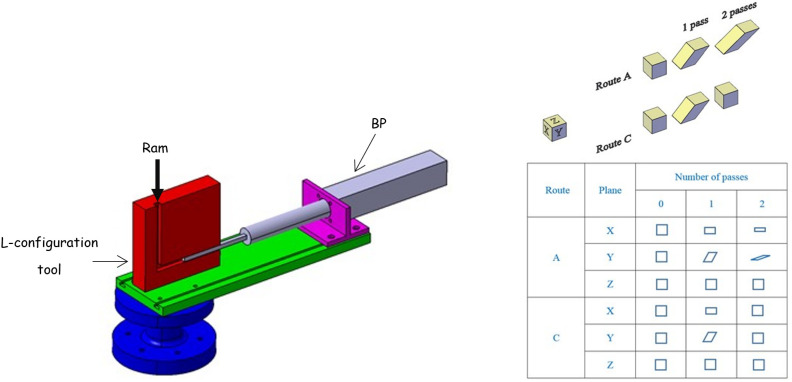
Schematic illustration of the L-configuration ECAE facility (90° die-angle tooling) equipped with BP and deformation after one extrusion and two extrusions via routes A and C (X: Transverse plane, Y: Flow plane, Z: Longitudinal plane).

**Figure 2 polymers-14-05287-f002:**
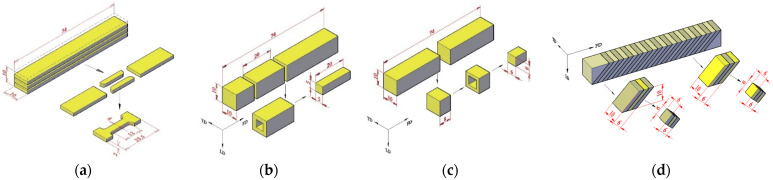
Specimen extraction for (**a**) tension tests, (**b**) shear tests and compression tests (**c**) for a piece with homogeneous deformation and (**d**) for a piece with periodic shear banding.

**Figure 3 polymers-14-05287-f003:**
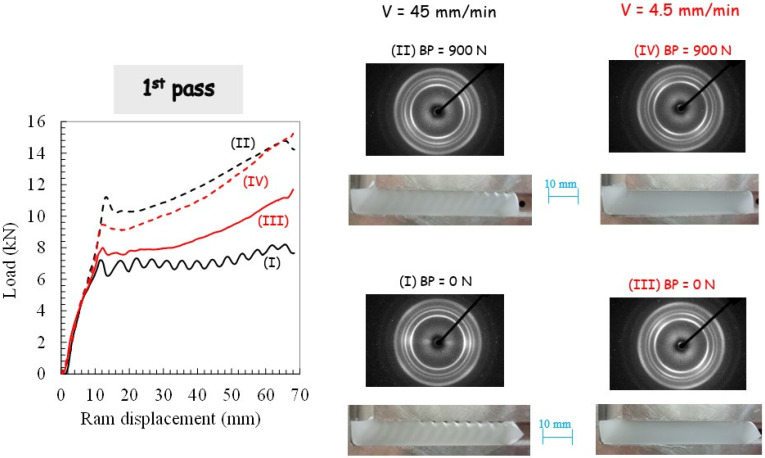
Results of the first pass extrusion in terms of load-displacement curve, deformation behavior and X-ray diffraction for different processing parameters: (**I**) Ram velocity of 45 mm/min and BP of 0 N, (**II**) Ram velocity of 45 mm/min and BP of 900 N, (**III**) Ram velocity of 4.5 mm/min and BP of 0 N, (**IV**) Ram velocity of 4.5 mm/min and BP of 900 N.

**Figure 4 polymers-14-05287-f004:**
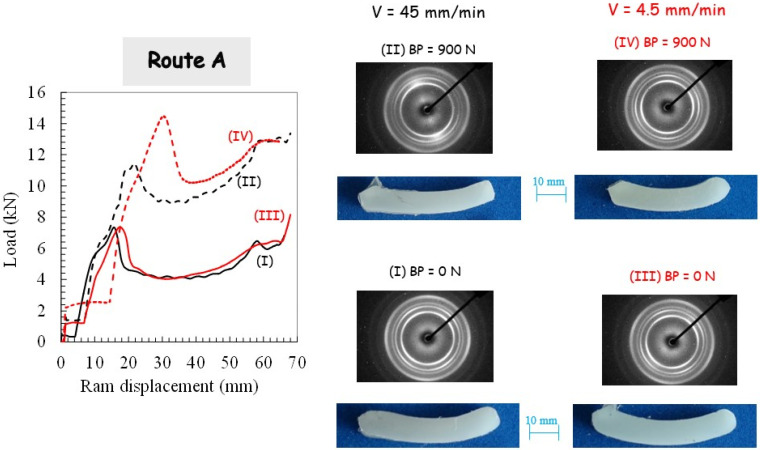
Results of the second pass extrusion in route A in terms of load-displacement curve, deformation behavior and X-ray diffraction for different processing parameters: (**I**) Ram velocity of 45 mm/min and BP of 0 N, (**II**) Ram velocity of 45 mm/min and BP of 900 N, (**III**) Ram velocity of 4.5 mm/min and BP of 0 N, (**IV**) Ram velocity of 4.5 mm/min and BP of 900 N.

**Figure 5 polymers-14-05287-f005:**
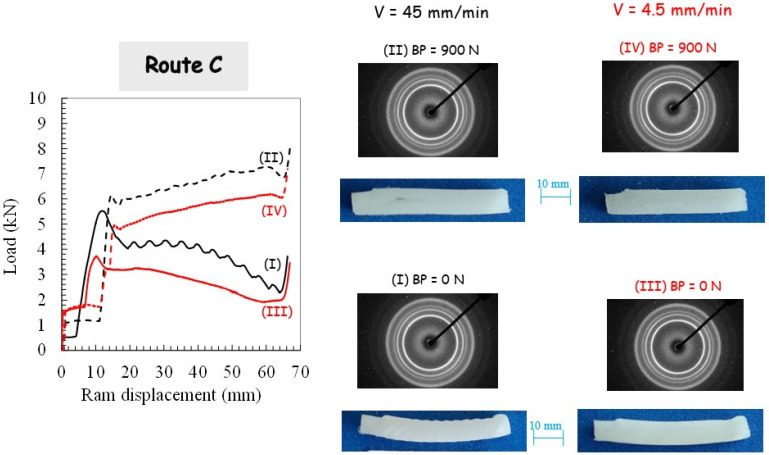
Results of the second pass extrusion in route C in terms of the load-displacement curve, deformation behavior and X-ray diffraction for different processing parameters: (**I**) Ram velocity of 45 mm/min and BP of 0 N, (**II**) Ram velocity of 45 mm/min and BP of 900 N, (**III**) Ram velocity of 4.5 mm/min and BP of 0 N, and (**IV**) Ram velocity of 4.5 mm/min and BP of 900 N.

**Figure 6 polymers-14-05287-f006:**
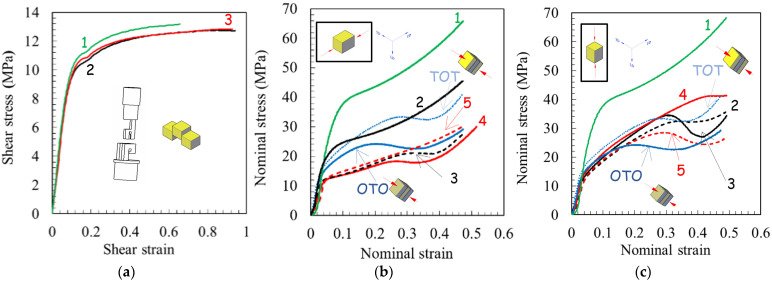
Shear and compressive results after first pass extrusion: (**a**) shear loading, (**b**) compression loading normal to longitudinal plane, and (**c**) compression loading normal to flow plane. One: as-received PP. Two: ram velocity of 45 mm/min and BP of 0 N. Three: ram velocity of 45 mm/min and BP of 900 N. Four: ram velocity of 4.5 mm/min and BP of 0 N. Five: ram velocity of 4.5 mm/min and BP of 900 N. OTO and TOT at ram velocity of 45 mm/min and BP of 0 N.

**Figure 7 polymers-14-05287-f007:**
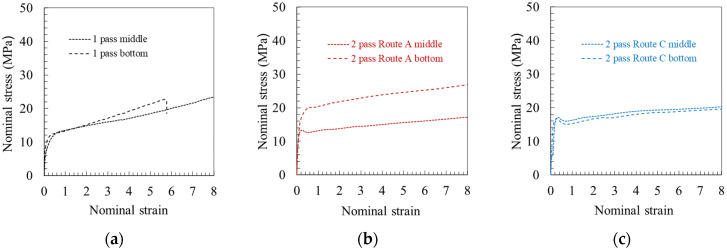
Tensile stress-strain curve for two specimens extracted from the middle and bottom parts of the extruded piece: (**a**) after first pass extrusion, (**b**) second pass extrusion in route A and (**c**) in route C (ram velocity of 45 mm/min and BP of 0 N).

**Figure 8 polymers-14-05287-f008:**
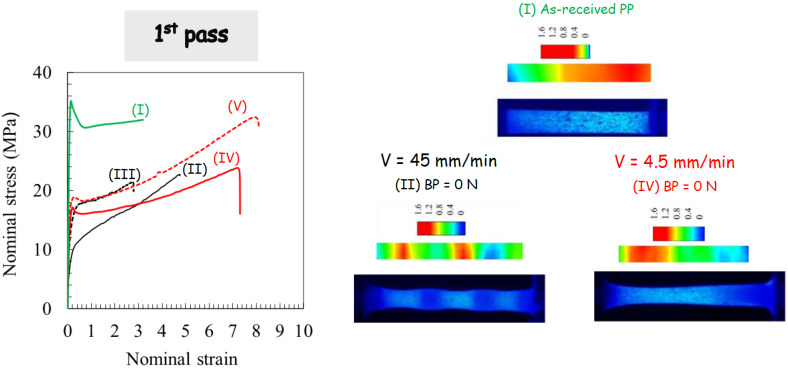
Tensile results after first pass extrusion in terms of stress-strain curve, deformation behavior and strain localization for different processing parameters: (**I**) as-received material, (**II**) ram velocity of 45 mm/min and BP of 0 N, (**III**) ram velocity of 45 mm/min and BP of 900 N, (**IV**) ram velocity of 4.5 mm/min and BP of 0 N, and (**V**) ram velocity of 4.5 mm/min and BP of 900 N.

**Figure 9 polymers-14-05287-f009:**
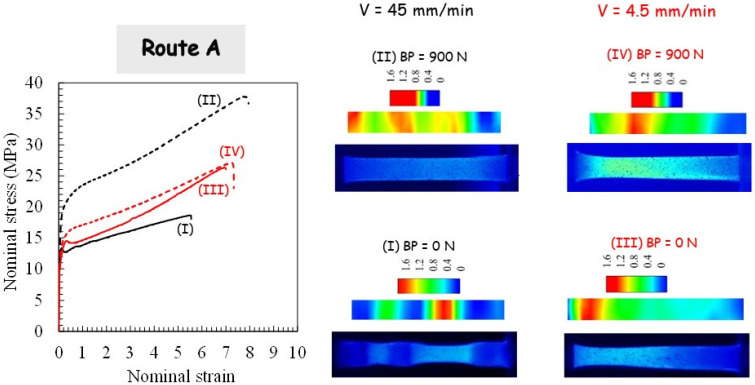
Tensile results after second pass extrusion in route A in terms of stress-strain curve, deformation behavior and strain localization for different processing parameters: (**I**) ram velocity of 45 mm/min and BP of 0 N, (**II**) ram velocity of 45 mm/min and BP of 900 N, (**III**) ram velocity of 4.5 mm/min and BP of 0 N, and (**IV**) ram velocity of 4.5 mm/min and BP of 900 N.

**Figure 10 polymers-14-05287-f010:**
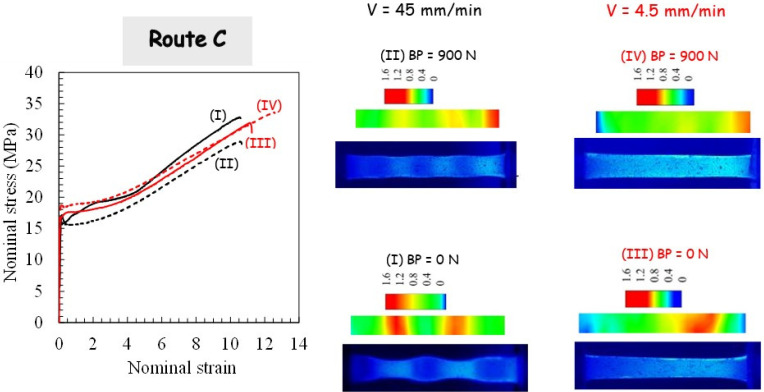
Tensile results after second pass extrusion in route C in terms of stress-strain curve, deformation behavior and strain localization for different processing parameters: (**I**) ram velocity of 45 mm/min and BP of 0 N, (**II**) ram velocity of 45 mm/min and BP of 900 N, (**III**) ram velocity of 4.5 mm/min and BP of 0 N, and (**IV**) ram velocity of 4.5 mm/min and BP of 900 N.

**Figure 11 polymers-14-05287-f011:**
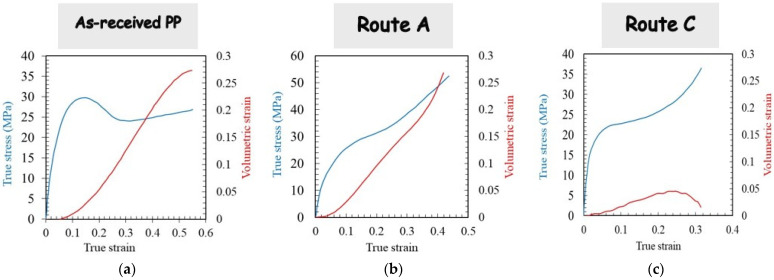
Tensile results in terms of stress and volumetric strain of (**a**) as-received material and after second pass extrusion (**b**) in route A and (**c**) in route C (ram velocity of 4.5 mm/min and BP of 0 N).

## Data Availability

The data presented in this study are available on request from the corresponding author.

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
