# Peer review of "Microstructure and Mechanical Properties of Severely Deformed Polypropylene in ECAE (Equal Channel Angular Extrusion) via Routes A and C"

_polymers, 2022, doi:10.3390/polym14235287_

Round 1

Reviewer 1 Report

Authors have already published article on this topic

https://www.sciencedirect.com/science/article/pii/S2352492820327653

How is this different

I doubt that the same paper is polished and submitted

Authors need to clarify

Author Response

The basic scientific content of our previous study (Ref. 29) has been yet understanding how ECAE process of PP works for producing high plastic deformation, when repeated until eight times, and its impact on tensile mechanical properties in relation to crystalline texturing.

The present paper is more particularly focused on the mechanism of occurrence of heterogeneous plastic deformations (periodic shear banding and warping) for PP specimens extruded twice via the processing routes A and C and, on the control and inhibition of the plastic instabilities via regulated back-pressure and ram velocity.

We would like to insist on the originality of the present paper that must be regarded as a companion paper of our previous study (Ref. 29).

Some valuable originalities ought to be mentioned. The changes of texturing that results from applying back-pressure and/or decreasing ram velocity are presented in the present paper Moreover, contrary to our previous study (Ref. 29), the present paper provides an analysis of the mechanical behavior of the extruded sample under a variety of conventional tests using different geometries and loadings, namely uniaxial tension, simple shear and uniaxial compression (considering the periodic shear banding if any). Another major added contribution relies on the analysis of the tensile deformation mechanisms of the extruded sample that were found altered by the processing route passing from a dominant cavitational mechanism for route A to a dominant shear yielding mechanism for route C.

We have revised the manuscript in order to highlight more clearly the added contribution of this work.

Reviewer 2 Report

Dear Author,

In this research article, you have studied the changes in the microstructure and mechanical properties of polypropylene using the ECAE method, which is one of the SPD techniques. This method is commonly used in the last two decades to study changes in the microstructure and mechanical properties of metals and alloys. You have studied polypropylene extrusion by routes A and C in a 90° die angle tool with back-pressure at different speeds and only two routes. You have studied the evolution of the structure using the wide-angle X-ray scattering technique. Unfortunately, I guess you haven't studied the microstructure with optical and scanning microscopes. For mechanical property changes, you have only studied uniaxial stress/compression and simple shear. At the same time, you showed the total displacements of the fields, converted into volume deformations, and the influence of deformation mechanisms on the tensile properties of elastic-plastic polypropylene. Unfortunately, you do little research on the effect of cyclic stress-compression tests at constant strain amplitude on the development of material physical-chemical properties. In the future, I recommend that this elastic-plastic material be tested using the so-called hard cyclic viscoplastic deformation (HCVD) test method to study the change of physical and chemical properties of polypropylene depending on the increase of accumulated deformation.

Author Response

We thank the reviewer for his suggestion. In the future, other solid state forming process should be employed for different polymer systems.

We would like to thank the reviewers for all precious comments, enabling us to enhance the content and the text of the paper. According to the reviewers’ remarks, modifications have been included into the manuscript.